# Cryo-EM structure of transcription termination factor Rho from *Mycobacterium tuberculosis* reveals bicyclomycin resistance mechanism

Emmanuel Saridakis[1,2,7], Rishi Vishwakarma[3,6,7], Josephine Lai-Kee-Him[3], Kevin Martin[3], Isabelle Simon[4,5], Martin Cohen-Gonsaud[3], Franck Coste [5], Patrick Bron [3✉], Emmanuel Margeat [3✉] & Marc Boudvillain [2,5✉]

The bacterial Rho factor is a ring-shaped motor triggering genome-wide transcription termination and R-loop dissociation. Rho is essential in many species, including in *Mycobacterium tuberculosis* where *rho* gene inactivation leads to rapid death. Yet, the *M. tuberculosis* Rho [Mtb Rho] factor displays poor NTPase and helicase activities, and resistance to the natural Rho inhibitor bicyclomycin [BCM] that remain unexplained. To address these issues, we solved the cryo-EM structure of Mtb Rho at 3.3 Å resolution. The Mtb Rho hexamer is poised into a pre-catalytic, open-ring state wherein specific contacts stabilize ATP in intersubunit ATPase pockets, thereby explaining the cofactor preference of Mtb Rho. We reveal a leucine-to-methionine substitution that creates a steric bulk in BCM binding cavities near the positions of ATP γ-phosphates, and confers resistance to BCM at the expense of motor efficiency. Our work contributes to explain the unusual features of Mtb Rho and provides a framework for future antibiotic development.

[1] Institute of Nanoscience and Nanotechnology, NCSR "Demokritos", Ag. Paraskevi, 15310 Athens, Greece. [2] Le Studium Loire Valley Institute for Advanced Studies, Orléans, France. [3] CBS (Centre de Biologie Structurale), Univ Montpellier, CNRS, INSERM, Montpellier, France. [4] ED 549, Santé, Sciences Biologiques & Chimie du Vivant, Université d'Orléans, Orléans, France. [5] Centre de Biophysique Moléculaire, CNRS UPR4301, rue Charles Sadron,  affiliated with Université d'Orléans, 45071 Orléans, cedex 2, France. [6] Present address: Department of Biochemistry and Molecular Biology, The Pennsylvania State University, University Park, State College, PA 16802, USA. [7] These authors contributed equally: Emmanuel Saridakis, Rishi Vishwakarma.
✉email: patrick.bron@cbs.cnrs.fr; margeat@cbs.cnrs.fr; marc.boudvillain@cnrs.fr

Tuberculosis is a major global health concern, each year killing ~1.5 million people worldwide. Multi-resistant strains of *Mycobacterium tuberculosis*—the causative agent of tuberculosis—arise at an alarming rate and there is an urgent need to better understand the mechanisms of drug resistance and to develop alternative therapeutic strategies against *M. tuberculosis*[1].

Transcription termination factor Rho is a central component of gene regulation in bacteria. It is essential in many Gram-negative species and in high G + C Gram-positive Actinobacteria such as *M. tuberculosis*[2] or *Micrococcus luteus*[3]. Since Rho has no structural homologs in eukaryotes, it is an attractive target candidate for the development of new antibiotics[4].

Some Actinobacteria of the *Streptomyces* genus produce a natural Rho inhibitor called Bicyclomycin [BCM]. BCM is effective against various Gram-negative pathogens but is inactive against most Gram-positive species, including *M. tuberculosis*[5]. One notable exception is *M. luteus*, whose growth is inhibited by BCM[3]. Accordingly, BCM strongly inhibits the in vitro enzymatic activity of *M. luteus* Rho ($_{Mic}$Rho)[3] but hardly affects *M. tuberculosis* Rho ($_{Mtb}$Rho)[6]. There is currently no rational explanation for this difference.

Our understanding of the mechanisms of Rho-dependent transcription termination (RDTT) and inhibition by BCM stems mostly from studies of *Escherichia coli* Rho ($_{Ec}$Rho). The $_{Ec}$Rho prototype is a ring-shaped, hexameric protein motor that dissociates transcription elongation complexes (TECs) in an RNA- and ATP-dependent manner[7–9]. Activation of $_{Ec}$Rho is triggered by binding to a C > G sequence-skewed and poorly structured *Rut* (Rho utilization) site in the nascent transcript and by allosteric closure of the $_{Ec}$Rho ring around the RNA chain[7–10]. Once activated, the $_{Ec}$Rho ring can hydrolyze ATP, translocate RNA, unwind RNA:DNA duplexes, and disrupt TECs[11–14]. Recent work supports a model where the $_{Ec}$Rho motor first binds RNA polymerase (RNAP) and allosterically destabilizes the TEC from this sitting position once it senses a *Rut* site in the emerging transcript[15–17].

The six N-terminal domains (NTDs) of $_{Ec}$Rho form a crown-like primary binding site (PBS) on one face of the hexamer ring[18]. This composite PBS includes a YC-binding pocket (Y being a C or U residue) on each $_{Ec}$Rho protomer that contributes to the specific recognition of transcript *Rut* sites[18,19]. The NTD also carries the residues contacting RNAP identified in recent cryoEM structures of the $_{Ec}$Rho:RNAP complex[15,16].

Despite these important roles, the NTD is not highly conserved and often contains large N-terminal insertion domains (NIDs), in particular in high G + C Actinobacteria[4]. In $_{Mic}$Rho and $_{Mtb}$Rho, these inserts increase the affinity for RNA, allow productive interaction with structured transcripts, and promote RDTT at promoter-proximal sites that the NID-less $_{Ec}$Rho is unable to use[6,20]. Indels also interrupt the PBS sequences of $_{Mic}$Rho and $_{Mtb}$Rho, which may be further indications of a *Rut*/RNA sensing mechanism deviating from the $_{Ec}$Rho paradigm.

The much more conserved C-terminal domain (CTD) of $_{Ec}$Rho is responsible for intersubunit cohesion and ATP-dependent RNA translocation[10,21]. The CTD notably carries the Walker A/B motifs forming ATPase pockets at subunit interfaces and the catalytic Glu, Arg valve, and Arg finger residues required for catalysis of ATP hydrolysis. These residues are highly conserved in phylo-divergent Rho factors, including in $_{Mtb}$Rho[4].

The CTD also contains the secondary binding site (SBS) Q- and R-loop motifs that translocate RNA through the $_{Ec}$Rho ring as a function of the chemical state of the ATPase pockets[10,21]. A K→T mutation in the R-loop (Lys326 in $_{Ec}$Rho to Thr501 in $_{Mtb}$Rho) slightly weakens enzymatic activity[6]. The CTD also carries the side chains that form an interaction pocket for BCM near each ATP binding site[22]. The binding of BCM to these pockets locks $_{Ec}$Rho in the open ring conformation, thereby preventing enzymatic activation[23]. The BCM-binding side-chains are strictly conserved in $_{Mtb}$Rho[4,6] and thus cannot account for its resistance to BCM.

To elucidate the origin of this resistance to BCM and to better comprehend the evolutionary specifics of RDTT in *M. tuberculosis*, we solved the structure of $_{Mtb}$Rho using cryo-EM single-particle analysis. Initial attempts by X-ray crystallography proved fruitless (see Supplementary Information) while cryo-EM led to a 3.3 Å resolution map. We show that $_{Mtb}$Rho can adopt an open, ring-shaped hexamer conformation that mimics that observed for $_{Ec}$Rho[18]. The NIDs are not resolved in the $_{Mtb}$Rho hexamer structure, consistent with predicted intrinsically disordered features. We identify a leucine-to-methionine substitution in $_{Mtb}$Rho that creates a steric bulk in the cavity where BCM normally binds. We show that this Leu→Met mutation is a taxa-specific evolutionary feature that alone is sufficient to account for $_{Mtb}$Rho resistance to BCM.

## Results and discussion

**High-resolution reconstruction of the $_{Mtb}$Rho complex**. The sequence and main motifs of $_{Mtb}$Rho are detailed in Supplementary Fig. 1. Freshly purified $_{Mtb}$Rho mostly forms monodisperse hexamers in presence of Mg-ATP and dC$_{20}$ ligands, as determined from SEC-MALS and SDS-PAGE experiments (Supplementary Fig. 2a, b). Images of negatively stained $_{Mtb}$Rho complexes revealed, from 2D class averages, a hexameric organization of particles forming closed rings (Supplementary Fig. 2c) while these rings appear mostly open in conventional cryo-EM images (Supplementary Fig. 2d). Since some particles display an open conformation and the intensity of the $_{Mtb}$Rho subunits is not uniform in negative stain images, we surmise that this discrepancy stems from a combination of several parameters such as the collapse and flattening of particles due to their drying during sample preparation, a preferential orientation of the negatively stained particles onto the carbon film, and a misalignment of hexameric particles during the 2D classification.

The most suitable conditions for high-resolution image acquisition in terms of $_{Mtb}$Rho particles distribution and orientation in ice in presence of Mg-ATP and dC$_{20}$ ligands were obtained using Lacey grids. We recorded 10,888 movies of the $_{Mtb}$Rho complexes using a Titan Krios instrument and processed them as described in Supplementary Figs. 3, 4. This analysis revealed that the overall organization of $_{Mtb}$Rho particles is similar to that of $_{Ec}$Rho and that 79 and 21% of particles respectively correspond to open and closed ring organizations of the complex. Further analysis of the set of particles classified as closed ring revealed that about 92,000 particles with an open conformation were still present in this dataset. These misaligned particles, which do not display well-defined structural features, were not included in the final dataset. This, of course, means that the overall percentage of $_{Mtb}$Rho particles truly in the closed ring conformation is even lower (16%). Moreover, the analysis of the closed ring dataset did not permit to compute a reliable and interpretable map, while particles corresponding to the open conformation allowed us to compute a cryo-EM map at 3.3 Å resolution (which led to the structure of the hexamer in the open ring conformation), showing an open-ring organization of the complex resulting from the assembly of six $_{Mtb}$Rho subunits (Fig. 1 and Supplementary Figs. 5–7). Local resolution mapped on the structure with the RELION package ranges from 3.1 to 4.8 Å (Fig. 1a–c). A gradient of high-to-low resolution is apparent in the 3D map from the innermost $_{Mtb}$Rho ring subunits (labeled with stars in Fig. 1b) to subunits at the ring gap (labeled with black dots). This suggests some degree of flexibility among

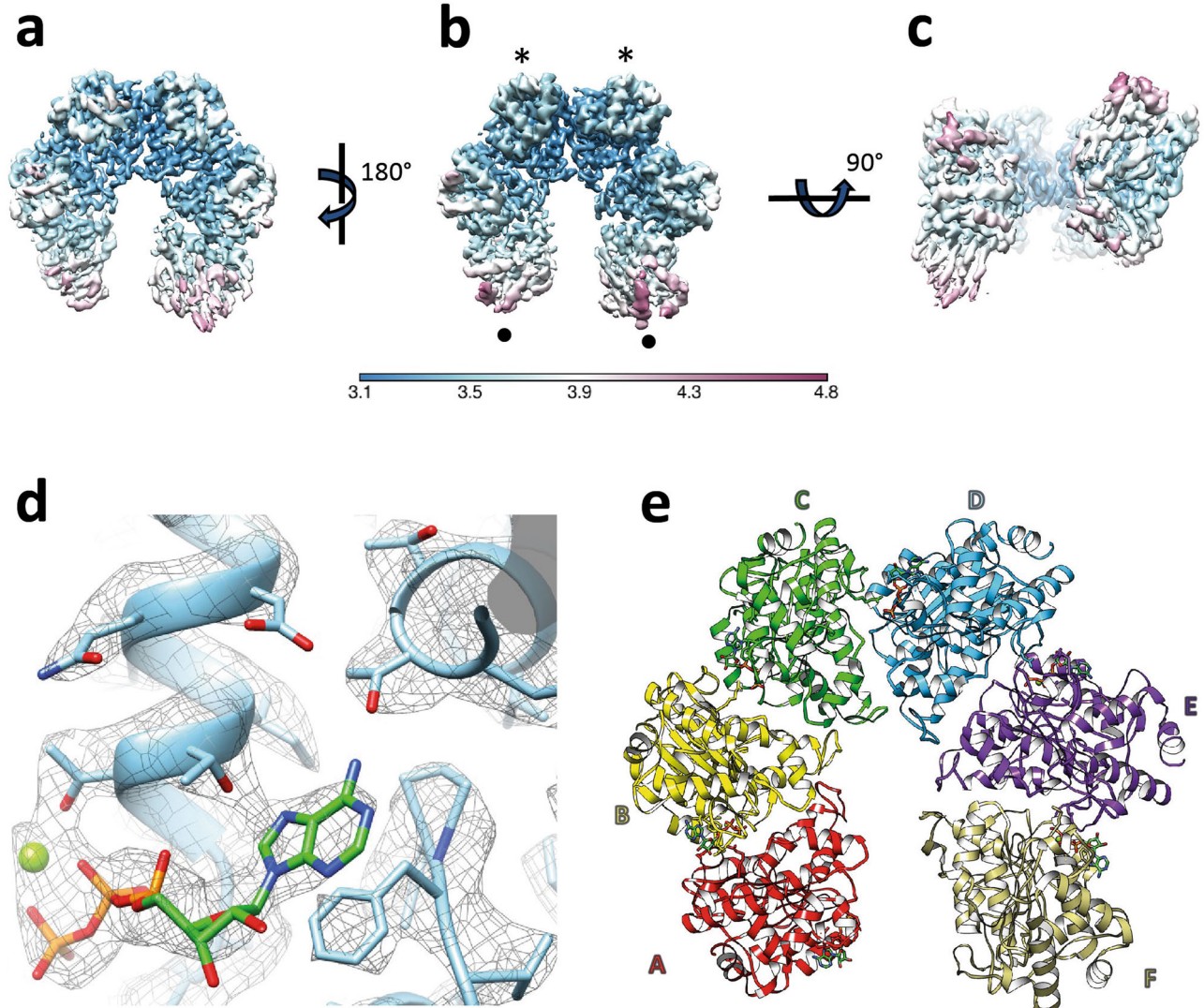

**Fig. 1 Single-particle cryo-EM structure of the *M. tuberculosis* Rho complex. a–c** The 3.3 Å cryo-map of _Mtb_Rho colored according to local resolution from highest (dark blue) to lowest resolution (dark purple). The _Mtb_Rho is composed of six subunits adopting an open corkscrew conformation. When observed from top (**a**) or down (**b**) views, _Mtb_Rho displays a U-like shape where the two subunits at the gap are the less well-resolved (black dots) while the innermost subunits (black stars) are the best resolved. The side view **c** highlights the shift between the two gap subunits that results from the corkscrew arrangement of the _Mtb_Rho ring. The first ~220 N-terminal residues of each protomer are not resolved. **d** Close-up view of the ATP binding site at the C/D interface. The _Mtb_Rho atomic model and cryo-EM map are superimposed. The ATP phosphates are in orange, sugar in green, oxygen groups in red and adenine base in green/blue. The catalytic $Mg^{2+}$ ion is shown as a green sphere. **e** Refined PDB structure of _Mtb_Rho. The subunits are labeled from A to F in clockwise order, observed from the CTD face of the _Mtb_Rho hexamer as in B.

subunits, which was confirmed by a multibody analysis where the first main eigenvectors correspond to intersubunit twists (Supplementary Fig. 8). An atomic model of the _Mtb_Rho complex was built and refined based on the cryo-EM density map (Table 1, Fig. 1, and Supplementary Fig. 9) and is detailed below. Although the map resolution is not uniform, all ATP binding sites are resolved and the _Mtb_Rho atomic structure fits well, even into the less well-resolved protomers of the cryo-EM map (Fig. 1d and Supplementary Figs. 5–8).

**The _Mtb_Rho hexamer adopts an open ring conformation.** The _Mtb_Rho hexamer adopts an open, corkscrewed configuration (Fig. 1) similar to the ones observed for _Ec_Rho (Supplementary Fig. 10) alone[18] or in complex with _Ec_RNAP[15,16]. An open, corkscrewed hexamer was also proposed for RNA-free Rho from *Thermotoga maritima*[24], supporting that this is a conserved and mechanistically important trait of the termination factor.

The N-termini, including NIDs, are not resolved in the structure of the _Mtb_Rho hexamer (Fig. 1). This is consistent with the intrinsically disordered state of the NID as predicted by the XtalPred server[25]. The $dC_{20}$ ligand is also not resolved despite its ability to bind and stabilize _Mtb_Rho, as determined by thermal shift assay[26]. It might be that $dC_{20}$ interacts with NTD (or NID) but is not long enough to induce sufficient folding for the domain to be resolved. Alternatively, the NTD/NID may properly fold upon $dC_{20}$ binding but adopt variable orientations with respect to the CTD due to a flexible NTD-CTD junction (Fig. 2a). In this case, the $dC_{20}$ ligand might be too short to bridge several NTDs/NIDs and restrict their movements. It is also possible that the NIDs substitute for the lineage-specific motifs that stabilize the _Ec_(Rho:RNAP) complex[15,16] and become structurally organized only upon _Mtb_Rho interaction with RNAP[27].

PBS Glu280 and indel2 residues (Supplementary Fig. 1) are also not resolved while the other PBS side-chains adopt somewhat

**Table 1 Cryo-EM data collection, refinement, and validation statistics.**

| | #1 name (EMDB-12701) (PDB 7OQH) |
|---|---|
| Data collection and processing | |
| Magnification | 165,000 |
| Voltage (kV) | 300 |
| Electron exposure (e−/Å$^2$) | 48.07 |
| Defocus range (μm) | −0.8 to −2 |
| Pixel size (Å) | 0.8141 |
| Symmetry imposed | C1 |
| Initial particle images (no.) | 2,217,252 |
| Final particle images (no.) | 986,385 |
| Map resolution (Å) | 3.32 |
| FSC threshold | 0.142 |
| Map resolution range (Å) | 3.1–4.7 |
| Refinement | |
| Initial model used (PDB code) | 1PVO |
| Model resolution (Å) | 3.3 |
| FSC threshold | 0.143 |
| Model resolution range (Å) | 3.1–4.7 |
| Map sharpening *B* factor (Å$^2$) | −107.639 |
| *Model composition* | |
| Non-hydrogen atoms | 16,824 |
| Protein residues | 2,169 |
| Ligand atoms (ATP) | 186 (6 ATP) |
| Ligand atoms (Mg$^{2+}$) | 5 |
| *B* factors (Å$^2$) | |
| Protein | 36.52 |
| Ligand (ATP) | 38.28 |
| Ligand (Mg$^{2+}$) | 28.89 |
| R.m.s. deviations | |
| Bond lengths (Å) | 0.004 |
| Bond angles (°) | 0.616 |
| Validation | |
| MolProbity score | 2 |
| Clashscore | 11.3 |
| Poor rotamers (%) | 0.3 |
| Ramachandran plot | |
| Favored (%) | 93.47 |
| Allowed (%) | 6.49 |
| Disallowed (%) | 0.05 |

variable orientations in the $_{Mtb}$Rho protomers (Fig. 2a), possibly because of the lack of stabilizing nucleic acid ligand. As a result, the PBS pockets do not seem as deep or as defined as in $_{Ec}$Rho[18], particularly at the level of the 5′C-binding subsite (Fig. 2b). Unfortunately, these structural features do not provide clues as to why $_{Mtb}$Rho, but not $_{Ec}$Rho, can bind and utilize structured RNA substrates efficiently[6,27,28]. Further work, e.g., with longer nucleic acid ligands, will be needed to address this specific question.

**Organization of the ATP-dependent allosteric network in $_{Mtb}$Rho.** The presence of a clear electron density in all the ATPase pockets of $_{Mtb}$Rho reveals that ATP ligands are held by a network of interactions involving the phosphate, sugar, and adenine base moieties (Figs. 1d, 2a, c). The interaction network also includes a catalytic Mg$^{2+}$ ion coordinating β and γ-phosphate oxygens (Figs. 1d, 2c). The adenine base is stacked on Phe530 while Thr333 and Thr361 form hydrogen bonds with the NH$_2$ and N$_7$ groups of adenine, respectively (Fig. 2c). These H-bonds explain the marked preference of $_{Mtb}$Rho for purine versus pyrimidine triphosphates[6]. Purine-specific interactions are not found in $_{Ec}$Rho, which hydrolyzes the four NTPs with comparable efficiencies[29]. In this case, the NTP base is held in sandwich by an aromatic contact with $_{Ec}$Phe355 (the equivalent of $_{Mtb}$Phe530) and a methionine-aromatic contact with $_{Ec}$Met186 (replaced by

Thr361 in $_{Mtb}$Rho) while $_{Ec}$Thr158 (the equivalent of $_{Mtb}$Thr333) is too far away (>3.5 Å in all $_{Ec}$Rho structures) to contribute to the interaction (Supplementary Fig. 11)[10,18,21].

Consistent with the open configuration of the $_{Mtb}$Rho ring (Fig. 1) and the absence of activating RNA ligand, the ATPase pockets are in an unproductive state. A direct interaction with the arginine finger (R541) from the adjacent protomer somewhat shields the γ-phosphate of ATP (Fig. 2c and Supplementary Fig. 11). Furthermore, the arginine valve (R387) and catalytic glutamate (E386) residues are too distant from, respectively, the γ-phosphate and catalytic Mg$^{2+}$ ion to catalyze ATP hydrolysis (Fig. 2c and Supplementary Fig. 11). By comparison, structures of the closed $_{Ec}$Rho ring (i.e., the catalytically-competent state) display tighter subunit interfaces (Supplementary Fig. 11) where the catalytic Glu (E211) and Arg valve (R212) residues are adequately positioned to bind and polarize the catalytic Mg$^{2+}$ ion and catalytic water molecule, respectively[10,21]. Tight contacts between the $_{Ec}$Rho subunits also allow the establishment of the allosteric communication network connecting the ATPase pocket to the RNA-binding SBS in the hexamer central channel (Supplementary Fig. 12a)[10]. Although all the network residues are conserved in $_{Mtb}$Rho, the corkscrewed disposition of the subunits (Supplementary Fig. 10) prevents the formation of the network contacts (Supplementary Fig. 12a). The allosteric network is also not formed in the pre-catalytic open ring structure of $_{Ec}$Rho[18].

The R- and Q-loops of $_{Mtb}$Rho are connected by a H-bond between the hydroxyl group of Thr501 and the carbonyl of Gly462 (Supplementary Fig. 12b). Such an interaction is not observed in $_{Ec}$Rho where the Lys326 side-chain (the equivalent of $_{Mtb}$Thr501) instead extends into the central channel (Supplementary Fig. 12b) as part of the RNA-binding SBS[10,21]. The H-bonded $_{Mtb}$Thr501 side-chain cannot play this role in $_{Mtb}$Rho (Supplementary Fig. 12b), suggesting that the SBS is restricted to Q-loop residues and explaining why a Thr→Lys mutation at position 501 stimulates the enzymatic activity of $_{Mtb}$Rho[6].

**A single Leu→Met mutation creates a steric constraint in the BCM binding pocket of $_{Mtb}$Rho.** The totality of residues identified as directly interacting with BCM in the crystal structure of the $_{Ec}$Rho-BCM complex[22] are strictly conserved in $_{Mtb}$Rho (Supplementary Fig. 1)[4]. However, close inspection of the BCM-binding cavity in our $_{Mtb}$Rho structure revealed that it contains a methionine (Met495) instead of a leucine (Leu320) in $_{Ec}$Rho. This Leu→Met substitution creates a bulk in an otherwise structurally comparable cavity (Fig. 3a), which could penalize BCM binding by steric clash. The hindrance could be with BCM itself or with the $_{Mtb}$Lys359 side-chain from the Walker A motif (aka P-loop). In $_{Ec}$Rho, the corresponding lysine (Lys184) undertakes a conformational change upon BCM binding. This movement may be impaired by the bulky Met495 neighbor in $_{Mtb}$Rho (Supplementary Fig. 13a).

To test this steric clash hypothesis, we compared the transcription termination activities of WT $_{Mtb}$Rho and its M495L mutant derivative using *E. coli* RNAP and a DNA template encoding the Rho-dependent λtR1 terminator. With this in vitro heterologous system, WT $_{Mtb}$Rho triggers efficient transcription termination, starting at promoter-proximal sites along the DNA template (Fig. 3b), as described previously[6]. High concentrations of BCM are required to perturb the termination activity of WT $_{Mtb}$Rho (Fig. 3b)[6]. The M495L mutant is also a very efficient termination factor but its activity is readily inhibited by ~10 times lower concentrations of BCM (Fig. 3b, c). Under the same experimental conditions, the IC50 values previously measured for the BCM-sensitive WT $_{Ec}$Rho[27] and here for the $_{Mtb}$Rho M495L mutant (Fig. 3c) are similar (~40 and ~35 μM,

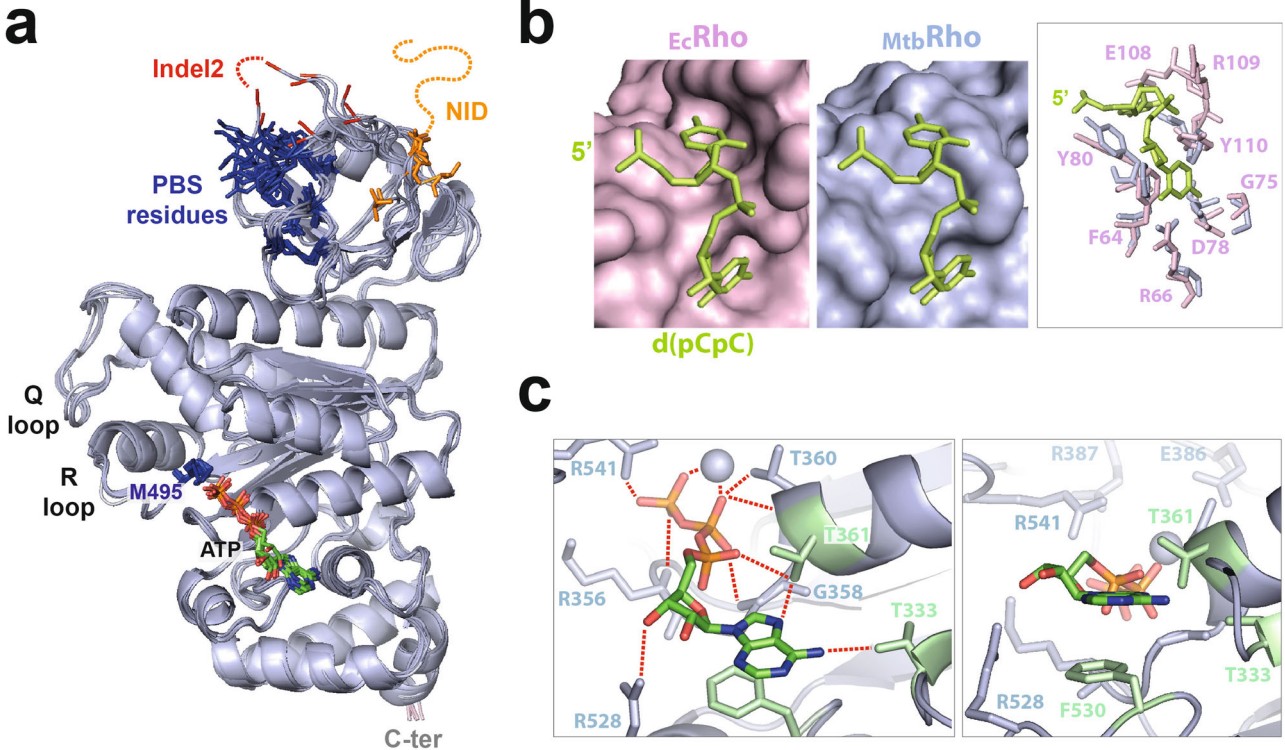

**Fig. 2 Organization of the ligands binding network in _Mtb_Rho. a** Superposition of the six _Mtb_Rho protomers with the main RNA and ATP binding motifs highlighted. ATP ligands are shown as in Fig. 1d with closeby Met495 side-chains in dark blue. Resolved residues equivalent to the PBS residues of _Ec_Rho are also shown in dark blue. **b** Comparison of the PBS pockets of _Ec_Rho and _Mtb_Rho. A dC$_2$ ligand (in green) has been modeled on _Mtb_Rho upon structural alignment with _Ec_Rho (PDB 1PV4). **c** Closeup views of the ATPase pocket at the best resolved interface (C/D) of _Mtb_Rho showing the network of interaction with the ATP ligand and catalytic Mg$^{2+}$ ion (sphere).

respectively). These data thus identify the Met495 side-chain as the main structural determinant of _Mtb_Rho resistance to BCM.

A Leu→Met mutation is found at the same position in the Rho sequences of various Actinobacteria of the Corynobacteriales order (Supplementary Fig. 14). This is notably the case for *Mycobacterium cholonae*, a bacterium that contains the cluster of genes responsible for BCM biosynthesis[30]. The Leu→Met mutation may thus render *M. cholonae* immune to its own BCM production. However, this protective mechanism does not appear to be widespread. Indeed, _Ec_Leu320 is strictly conserved in the Rho sequences of other bacteria bearing the BCM biosynthesis gene cluster (Supplementary Fig. 14), including the *Streptomyces* species known to produce BCM[31].

Apart from Actinobacteria, we retrieved the Leu→Met mutation in two closely related insect endosymbionts from the Gram-negative Bacteroidetes phylum (Supplementary Fig. 14). This finding is surprising given the endosymbiont lifestyle, which itself should provide protection against BCM exposure. Although this suggests that the Leu→Met mutation has been acquired by chance or for another (unknown) reason than resistance to BCM, it is not clear yet if this conjecture can be extended to the case of Actinobacteria.

Overall, the Leu→Met mutation is rare and the only side-chain change observed at this position within the ~1300 representative Rho sequences compiled previously from multiple taxa[4].

**Fitness cost associated to BCM resistance.** An unexpected effect of the M495L mutation is the stimulation of transcription termination when compared to WT _Mtb_Rho (~80% versus ~60% apparent efficiency in absence of BCM; Fig. 3c). Early termination is also favored with the M495L mutant (Fig. 3b, graph),

suggesting that the Met495 side-chain reduces the efficiency of the _Mtb_Rho motor.

To confirm this idea, we compared the helicase activities of M495L and WT _Mtb_Rho. As described previously, WT _Mtb_Rho is unable to unwind long RNA:DNA hybrids (Fig. 4a)[6]. By contrast, the M495L mutant readily unwinds a model RNA:DNA hybrid (57 base pairs), albeit at a slower rate than the _Ec_Rho control (Fig. 4a). This residual deficiency may be due to the R-loop Thr501 side-chain (versus Lys326 in _Ec_Rho), which also reduces the enzymatic efficiency of _Mtb_Rho[6]. Alternatively, The *Rut* site within the model RNA:DNA substrate may be more adapted to _Ec_Rho than to _Mtb_Rho requirements.

Notwithstanding, in the presence of an excess of the model poly[rC] substrate, the M495L mutant hydrolyses ATP at a steady-state rate similar to that measured for _Ec_Rho and about 3 times faster than WT _Mtb_Rho (Fig. 4b). Intriguingly, M495L hydrolyses the four rNTPs with comparable efficiencies (Fig. 4b) whereas WT _Mtb_Rho has a marked preference for purine vs pyrimidine triphosphates[6].

As mentioned above, pyrimidine triphosphates are probably less tightly bound in the ATPase pockets of _Mtb_Rho than in the pockets of _Ec_Rho due to a Met→Thr substitution (_Mtb_Thr361 vs _Ec_Met186) and loss of purine-specific contacts (Fig. 2c and Supplementary Fig. 11). We propound that in WT _Mtb_Rho, NTP hydrolysis is slowed by the Met495 side-chain because its greater-than-leucine bulk hinders changes in the allosteric network that connects the Walker motifs to the RNA-binding Q- and R-loops (Supplementary Figs. 12, 13). Poorly bound pyrimidine triphosphates may thus dissociate before hydrolysis whereas purine triphosphates are sufficiently stabilized by the network of _Mtb_Rho interactions to the base moiety (Fig. 2c). We speculate that steric hindrance is diminished by the M495L mutation to the extent

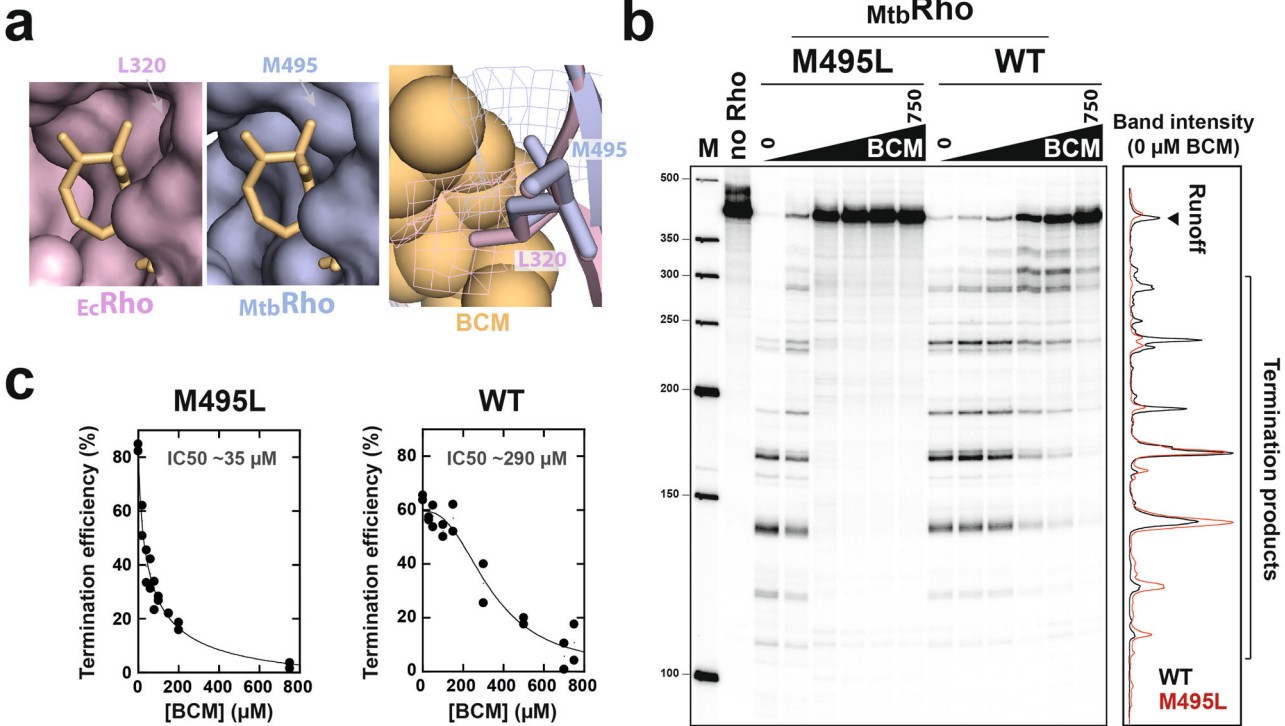

**Fig. 3 The $_{Mtb}$Met495 side-chain confers BCM resistance to $_{Mtb}$Rho. a** The BCM binding pockets of $_{Ec}$Rho (PDB 1PV4) and $_{Mtb}$Rho (this work). BCM has been fitted into the pockets by structural alignment with BCM-bound $_{Ec}$Rho (PDB 1XPO). $_{Ec}$Rho (left panel) or $_{Mtb}$Rho (middle panel) are shown alone in full surface representation. In the right panel, both structures are superimposed with the surfaces of the $_{Mtb}$M495 and $_{Ec}$L320 residues in mesh representation; BCM (sphere representation) only invades the mesh surface of $_{Mtb}$M495, illustrating a potential steric clash. **b** The M495L mutation sensitizes $_{Mtb}$Rho to BCM. A representative PAGE gel shows the effects of increasing concentrations of BCM (0–750 µM) on the transcription termination activities of the WT $_{Mtb}$Rho and M495L mutant. The graph compares the gel lane profiles for the WT (black) and M495L (red) proteins in absence of BCM (lanes 0). **c** IC50 graphs deduced from $n = 2$ independent transcription termination experiments.

that hydrolysis of pyrimidine triphosphates becomes faster than their dissociation from the ATPase pockets.

Taken together, these data illustrate how the resistance of $_{Mtb}$Rho to BCM has been acquired at the expense of enzymatic proficiency. Fitness cost is often associated to antibiotic resistance and has been observed previously for BCM-resistant *E. coli* strains bearing rare Rho mutations[32]. These observations and the otherwise very high level of phyletic conservation of the BCM binding cavity[4] support that the configuration of the cavity is substantially constrained by Rho function, thereby making it an attractive target for future drug development.

## Conclusion

Upon solving the first structure of a Rho factor from a Gram-positive bacterium (Fig. 1), we illustrated the general principles and lineage-specific variations that govern RDTT across the bacterial kingdom. Furthermore, we elucidated the molecular mechanism of resistance to bicyclomycin displayed by *M. tuberculosis* and, most likely, by Corynobacteriales relatives (Supplementary Fig. 14). This information may be used for future drug development using rational, structure-based approaches. Despite these notable achievements, further work will be needed to fully unravel the role(s) and mechanism of action of the poorly conserved, yet functionally important NID of $_{Mtb}$Rho. Identification of cognate RNA ligands will likely constitute a critical step towards this goal.

## Experimental procedures

**Materials.** Chemicals and enzymes were purchased from Sigma-Aldrich and New England Biolabs, respectively. Bicyclomycin

(BCM) was purchased from Santa Cruz Biotechnology. Radio-nucleotides were from PerkinElmer. DNA templates for in vitro transcription termination (Supplementary Table 1) were prepared by standard PCR procedures[33]. RNA strands for duplex unwinding assays (Supplementary Table 1) were obtained by in vitro transcription of PCR amplicons with T7 RNA polymerase and purified by 10% denaturing polyacrylamide gel electrophoresis (PAGE). Plasmid for overexpression of the M495L mutant was prepared by Quickchange (Stratagene) mutagenesis of the pET28b-MtbRho plasmid encoding WT $_{Mtb}$Rho (kindly provided by Dr. Rajan Sen, Hyderabad, India).

**Expression and purification of $_{Mtb}$Rho.** WT $_{Mtb}$Rho and the M495L mutant were overexpressed in Rosetta 2(DE3) cells (Merck-Millipore) harboring the appropriate pET28b derivative[34]. Cells were resuspended in lysis buffer (20 mM HEPES, pH 7.5, 300 mM NaCl, 2 mM β-mercaptoethanol, 5% glycerol) supplemented with an oComplete Protease Inhibitor tablet (Roche), 0.2 mg/mL lyso-zyme, and 0.05% sodium deoxycholate and incubated for 20 min at room temperature. Genomic DNA was broken by sonication with a Bioblock Vibra-Cell apparatus (10 s on/off cycles for 5 min at 30% amplitude). Crude protein lysates were fractionated by polymin-P (5%) and ammonium sulfate (0.5 g/mL) precipitations before purification by affinity chromatography on a HisTrap FF column (20 mM Tris-HCl, pH 7.6, 0.1 mM EDTA, 2 mM β-mercapt-toethanol, 5% glycerol, 10–500 mM imidazole gradient), cation exchange chromatography on a HiTrap SP sepharose (20 mM Tris-HCl, pH 7.6, 0.1 mM EDTA, 2 mM β-mercaptoethanol, 5% glycerol, 150–600 mM NaCl gradient), and gel filtration chroma-tography on a HiLoad 16/600 Superdex 200 column (20 mM

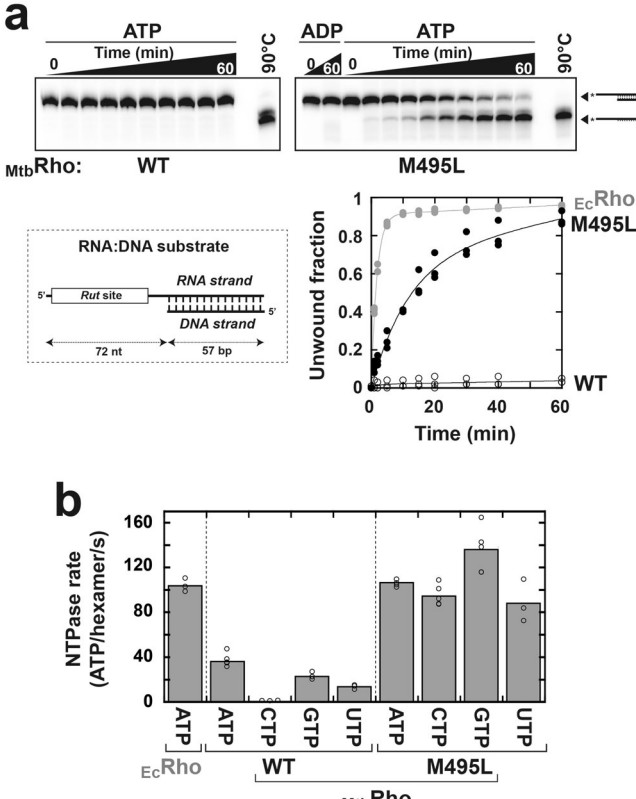

**Fig. 4 The Met495 side-chain impairs the activity of the ₘₜᵦRho motor.** The M495L mutation stimulates the RNA-DNA duplex unwinding (**a**) and NTP hydrolysis (**b**) activities of ₘₜᵦRho. A loss in NTP selectivity is observed for the M495L mutant as compared to WT ₘₜᵦRho. Graph and histogram values were deduced from $n = 3$ independent experiments except for ATPase with M495L ($n = 4$), ATPase with WT ₘₜᵦRho ($n = 5$), and CTPase and GTPase with M495L ($n = 5$). Bars in panel **b** correspond to mean values.

Tris-HCl, pH 7.9, 5% glycerol, 0.2 mM EDTA, 0.2 M KCl, 0.2 mM DTT[34]. All columns were from GE Healthcare. Purified ₘₜᵦRho proteins were used directly for cryo-EM or stored at −20 °C as micromolar hexamer solutions in storage buffer (100 mM KCl, 10 mM Tris-HCl, pH 7.9, 0.1 mM EDTA, 0.1 mM DTT, 50% glycerol) for biochemical assays.

**Cryo-EM sample preparation and data collection.** The ₘₜᵦRho-ATP-DNA complex sample was prepared by mixing 0.6 mg/mL of freshly purified WT ₘₜᵦRho with 10 μM dC₂₀ oligonucleotide and 1 mM ATP[26] in cryoEM buffer (150 mM NaCl, 5 mM MgCl₂, 10 mM Tris-HCl, pH 7.6). Three microliters of the mixture were applied to glow-discharged Lacey 300 mesh copper grids (Ted Pella Inc.), blotted for 3–4 s, and then flash-frozen in liquid ethane using the semi-automated plunge freezing device Vitrobot Mark IV (ThermoFisher Scientific) maintained at 100% relative humidity and 22 °C. Preliminary images were recorded using a JEOL 2200 FS electron microscope operating at 200 kV in zero-energy-loss mode with a slit width of 20 eV and equipped with a 4k × 4k slow-scan CCD camera (Gatan inc.). High-resolution data were collected at the EMBL cryo-EM core facility (Heidelberg, Germany) with a Titan Krios S-FEG transmission electron microscope (ThermoFisher Scientific) operating at 300 kV at a nominal magnification of X 165,000 under low-dose conditions with defocus values ranging from −0.8 to −2 μm, using the SerialEM automated acquisition software[35]. Movies were recorded with a K2-Summit direct electron detector (Gatan Inc.) configured in counting mode, mounted on a Gatan

Quantum 967 LS energy filter using a 20 eV slit width in zero-loss mode. The scheme of image recording was made with 7 images per position plus 1 middle, and as a series of 40 frames per movie, with 1.2 e⁻/Å² per frame, therefore giving a total accumulated dose of 48 e⁻/Å², with a corresponding pixel size of 0.81 Å/pix.

**Image processing and high-resolution 3D reconstruction.** A total of 10,888 movies were recorded. The frames of each movie were computationally dose-weighted and corrected for drift and beam-induced movement in RELION-3.1.0[36] using its own implementation. The contrast transfer function of each micrograph was determined using Gctf-v1.18 program[37]. A total of 2,217,252 particles were automatically picked with RELION's autopicking procedure. Briefly, a template-free autopicking procedure based on a Laplacian-of-Gaussian filter allowed to select and extract a set of particles from an initial small set of images and to compute corresponding 2D class averages. The best 2D class averages were then used as references in a second round of autopicking in order to optimize picking parameters. Once determined, the autopicking procedure was applied to all images. Particles were then sorted according to their correlation with the 2D class average used as reference, leading to discarding about 460,000 particles. The selected particle images were then extracted with a box size of 320 × 320 pixels, binned to obtain a pixel size of 3.25 Å/pix, and submitted to 2D classification. Using the dedicated Relion procedure, the best 2D class averages allowed computation of an ab initio model that was similar to the map computed from the atomic structure of the open form of ₑᵧRho (PDB 1PVO), confirming that the overall organization of ₘₜᵦRho was similar to that of ₑᵧRho. Consequently, we performed a 3D classification using as initial models two density maps computed from the open and closed conformation of ₑᵧRho (PDB 1PVO and 2HT1). This analysis revealed that 79 and 21% of particles corresponded to the open and closed conformation of ₘₜᵦRho, respectively. The particles corresponding to the open form of ₘₜᵦRho were subjected to a 3D classification with 4 classes using as initial model a map computed from the open ₑᵧRho structure. Two classes corresponding to 986,385 particles were selected and particles re-extracted with a box size of 400 × 400 pixels and binned at 1.49 Å/Pix. After two rounds of 3D auto-refinement without symmetry, a first density map at 3.97 Å was obtained. We then proceeded with per-particle defocus estimation, beam-tilt estimation plus Bayesian polishing. The final 3D refinement allowed us to compute a map at 3.32 Å resolution (FSC = 0.143) after post-processing. Figures were prepared using Chimera[38].

**Structure refinement.** The map coefficients were converted into structure factors using the corresponding Phenix cryo-EM tool[39], and molecular replacement was performed with Molrep[40], using one of the monomers of ₑᵧRho (chain C of PDB entry 1PV4) as search model. Four of the six chains were immediately placed and the remaining two (the outer ones of the open ring) were added in subsequent runs. The electron density corresponding to these outer chains (A and F) was overall weaker and less clear than that of the inner chains. The homohexameric structure was initially refined by Refmac[41] using the maximum-likelihood method, and subsequently with the Phenix real-space refinement tool[42]. Non-crystallographic symmetry, secondary structure, Ramachandran, and rotamer restraints were used. The refinement cycles were alternated with extensive manual model-building, fitting, and validation in Coot[43]. Six ATP molecules were fitted into clearly visible electron density, but no density imputable to nucleic acid could be identified. Additional density found near the β and γ-phosphates in five out of the six chains was assigned to catalytic Mg²⁺ ions coordinated to β and γ-phosphate oxygens. No

electron density corresponding to residues before Val222 and after Ser592 (before Val223 and after Val589 for chain F) was visible. Gaps in the density meant that residues after Lys277 and before Phe288 in chain C could not be modeled (corresponding residues for the other chains are A: 278–287, B: 276–285, D: 278–286, E: 276–286, F: 279–287). The final refinement run gave a model-to-map fit (CC_mask) of 0.7753, Molprobity all-atom clashscore of 11.32, 0.05% outliers in the Ramachandran plot (93.47% residues in favored and 6.49% in allowed geometries) and 0.3% sidechain outliers. The EMringer score of the final model calculated with Phenix[39] is 1.29.

**NTPase assays.** NTP hydrolysis activities were determined with a thin layer chromatography (TLC) assay[44]. For each NTP, a mixture of the cold NTP and matching $\alpha[^{32}P]$-NTP (or $\gamma[^{32}P]$-NTP) was used. Reaction mixtures contained 20 nM Rho hexamers, 1 mM of the NTP/$[^{32}P]$-NTP mixture, and 10 μM poly[rC] in NTPase buffer (50 mM KCl, 1 mM MgCl$_2$, 20 mM HEPES, pH 7.5, 0.1 mM EDTA, and 0.1 mM DTT) and were incubated at 37 °C. Reaction aliquots were withdrawn at various times, quenched with four volumes of 0.5 M EDTA, and stored on ice. Once all collected, the aliquots were spotted on a PEI-cellulose TLC plate (Sigma-Aldrich) and developed with 0.35 M potassium phosphate buffer (pH 7.5). Plates were dried and analyzed by phosphorimaging with a Typhoon FLA 9500 imager (GE Healthcare).

**Duplex unwinding assays.** RNA:DNA duplexes were prepared by mixing 5 pmoles of $^{32}P$-end labeled and 30 pmoles of unlabeled RNA strand with 60 pmoles of complementary DNA oligonucleotide (see Supplementary Table 1 for sequences) in helicase buffer (150 mM potassium acetate, 20 mM HEPES, pH 7.5, 0.1 mM EDTA). Mixtures were heated at 90 °C and slowly cooled to 20 °C. Duplexes were then purified by native 6% PAGE and stored in helicase buffer at −20 °C before use. Helicase reaction premixes[45] were assembled by mixing 5 nM $^{32}P$-labeled RNA:DNA duplex with 20 nM Rho hexamers in helicase buffer supplemented with 0.1 mg/mL BSA and incubated for 3 min at 30 °C. Then, helicase reactions were initiated by addition of a mix containing MgCl$_2$ and ATP (1 mM, final concentrations) and an excess of Trap oligonucleotide (400 nM final concentration) before further incubation at 30 °C. Reaction aliquots were taken at various times, mixed with two volumes of quench buffer (30 mM EDTA, 0.75% SDS, 150 mM sodium acetate, 6% Ficoll-400), and loaded on 8% PAGE gels containing 0.5% SDS. Gels were analyzed by Typhoon phosphorimaging[45] and ImageQuant TL software (Cytivia).

**Transcription termination experiments.** Transcription termination reactions[6,27,33] were assembled by mixing DNA template (0.1 pmol), E. coli RNAP (0.3 pmol), Rho (0 or 1.4 pmol), Superase-In (0.5 U/μL; Ambion) and BCM (0–15 nmol, Santa) in 18 μL of transcription buffer (40 mM Tris-HCl, pH 8, 5 mM MgCl$_2$, 1.5 mM DTT, and 100 mM KCl). Mixtures were incubated for 10 min at 37 °C before addition of 2 μL of initiation solution (250 μg/mL rifampicin, 2 mM ATP, GTP, and CTP, 0.2 mM UTP, and 2.5 μCi/μL $\alpha[^{32}P]UTP$ in transcription buffer). After 20 min of incubation at 37 °C, reactions were stopped upon addition of 4 μL of EDTA (0.5 M), 6 μL of tRNA (0.25 mg/mL), and 80 μL of sodium acetate (0.42 M) followed by ethanol precipitation. Reaction pellets were resuspended in loading buffer (95% formamide; 5 mM EDTA) and analyzed by denaturing 7% PAGE and Typhoon phosphorimaging. Transcript sizes were estimated by comparing gel band migrations with those of control RNA/DNA species and used for rough $\alpha[^{32}P]U$ content

normalization of band intensities[33]. The normalized band intensities were used to estimate apparent termination efficiencies (TE$_{app}$)[6,27]:

$$\mathrm{TE_{app}} = \frac{\sum I_{\mathrm{term}}}{\sum I_{\mathrm{term}} + I_{\mathrm{runoff}}} \times 100$$

where $\sum I_{\mathrm{term}}$ is the sum of the normalized intensities of the bands corresponding to termination products and $I_{\mathrm{runoff}}$ is the normalized intensity of the runoff product band.

IC$_{50}$ values for BCM inhibition were deduced by fitting the TE$_{app}$ data to the following equation:

$$\mathrm{TE_{app}} = (\mathrm{TE_{app}})_0 - F_{\max} \times \frac{[\mathrm{BCM}]^n}{[\mathrm{BCM}]^n + [\mathrm{IC_{50}}]^n}$$

where $(\mathrm{TE_{app}})_0$ is the value of TE$_{app}$ at 0 μM BCM, $F_{\max}$ is the maximal fraction of TE$_{app}$ that is sensitive to BCM inhibition, and $n$ is an empirical parameter that defines the sigmoid shape of inhibition[46].

**Statistics and reproducibility.** Transcription experiments were repeated independently twice and other biochemical experiments at least three times with similar results.

**Reporting summary.** Further information on research design is available in the Nature Research Reporting Summary linked to this article.

### Data availability
The cryo-EM data and the model were deposited as entries EMD-12701 and PDB 7OQH, respectively. The source data underlying Figs. 3c, 4a and 4b are provided as Supplementary Data 1. Uncropped gel scans can be found in Supplementary Data 1 while original Typhoon phosphorimager files (.gel format) are available at https://doi.org/10.5281/zenodo.5786227. All other data that support the findings of this study are available from the corresponding authors upon reasonable request.

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

## Acknowledgements

We warmly thank Annie Schwartz for her help with biochemical experiments. This work benefited from access to the cryo-EM facility of the European Molecular Biology Laboratory (EMBL) in Heidelberg with support from the iNEXT-Discovery program (project #871037) funded by the Horizon 2020 framework of the European Commission. The work was supported by grants from the French Agence Nationale de la Recherche (ANR-15-CE11–0024–01 to EM and ANR-15-CE11–0024–02 to MB), a sabbatical research fellowship from LE STUDIUM Loire Valley Institute for Advanced Studies (Marie Sklodowska-Curie 665790) to E.S., and a doctoral fellowship from Région Center-Val de Loire to I.S. CBS is a member of the French Infrastructure for Integrated Structural Biology (FRISBI) supported by Agence Nationale de la Recherche (ANR-10-INBS-05).

## Author contributions

R.V., J.L.K.H., and K.M. performed cryoEM experiments and image processing; E.S. and F.C. performed structure refinement; E.S., R.V., I.S., M.C.G., and F.C. performed protein preparation and biochemical experiments; P.B., E.M., and M.B. supervised the work and wrote the paper with help from other authors; all authors participated in data analysis.

## Competing interests

The authors declare no competing interests.
