## [Transparent Peer Review File · Communications Biology]

Reviewers' comments:

Reviewer #1 (Remarks to the Author):

Saridakis et al in their manuscript describe the cryo-EM structure of *M. tuberculosis* Rho (MtbRho) at 3.3 Å resolution. Although it lacks the information on the unique N-terminal insertion domain of the protein which makes it different from Rho protein from well studied systems, it provides valuable insights into its co-factor preference, poor helicase activity and purine preference for ATPase activity. The structural insights into MtbRho provide an explanation for lower susceptibility to bicyclomycin (BCM). They also suggest that this reduced resistance to BCM is compensated by reduced motor efficiency of the protein. This is a thorough structural analysis of essential function from an important pathogen, further delineating the individual contribution of key residues. The experiments are for the most part well designed and the conclusions are supported by the data. I have a few remarks which authors may find useful if they desire to undertake the revision of the manuscript.

The comparison of Mtb Rho structure with that of *E.coli* revealed that most of BCM interacting residues are conserved. They hypothesize that the Met495 in MtbRho creates a bulk that can affect BCM binding by the steric clash. In vitro transcription assays using heterologous *E.coli* system were used to test this hypothesis. Would this strategy capture the species-specific signatures of termination and its inhibition given the differences between the species and their Rho? M495L substitution in Mtb Rho converted it into more efficient ('*E.coli* like') using the heterologous transcription assay system. Unlike WT MtbRho, M495L substitution in the protein was able to unwind long RNA: DNA hybrids. As mentioned above an Mtb-based helicase activity assay system, using Mtb Rho-dependent terminator or RNA:DNA hybrids is likely to be more insightful given that Mtb Rho deals with high GC rich substrates.

I would suggest the usage of term resistance to BCM is changed to lower susceptibility. From the data they presented in Fig 3B and C, BCM inhibits MtbRho albeit at higher concentrations. Other groups working on Mtb Rho have also reported these results. Use of the word resistance simply adds confusion in the literature.

From the structural analysis, it is clear that M495 is the main structural determinant of Mtb Rho introducing steric clash. They argue that hindrance could be with BCM itself or with the MtbLys359 side-chain from the Walker A motif (aka P-loop). To address these, BCM binding could be tested.

V. Nagaraja

Reviewer #2 (Remarks to the Author):

Saridakis and colleagues report the first structure, obtained by cryo-EM, of the transcription termination factor Rho from a Gram-positive bacterium, namely *Mycobacterium tuberculosis*. This structure allows insights into the configuration of the Mtb-Rho hexamer, the organization of network required to bind ATP, and provided a hypothesis for the mechanism by which Mtb-Rho resists inactivation by bicyclomycin (BCM). This hypothesis, which predicted that a Leu/Met exchange at position 495 renders Mtb-Rho BCM resistant, was tested and validated biochemically. Interestingly, the M495L mutation causes BCM-resistance at the cost of enzymatic proficiency. Finally, the authors report on the evolutionary distribution of M495L-based BCM-resistance.

Rho is an attractive target for the development of new antibacterial agents, e.g. for the treatment of Tuberculosis. The lack of structural information on Rho from *M. tuberculosis* (or another gram-positive pathogen) has restricted these drug development efforts to somewhat unspecific phenotypic screen. This work thus addresses a major open question and I find it to be of high significance. It is unfortunate that the N-terminal insertion domain could not be resolved in this structure, but this is a minor weakness.

Overall, this work is of very high quality and my suggestions are all minor:

1. Please check the labels in figure 1A. At least in my version, they seem incorrect and possibly incomplete.
2. I find the discussion in the evolutionary tradeoff section somewhat ambiguous. Perhaps, it could be clarified if the authors want to suggest that the M495L mutation has been selected in mycobacteria specifically to mediate resistance to BCM. If so, it would be helpful to discuss what the evidence for this might be. Alternatively, the authors might want to add a few sentences on why this mutation occurs in mycobacteria if it was not selected to mediate BCM resistance.
3. I don't expect the authors to add data on this (unless they chose to), but it would of course be interesting to know if a BCM-susceptible mutant of Mtb-Rho can replace the wt copy of Rho (in Mtb or *M. smegmatis*) and if this replacement makes mycobacteria BCM-sensitive.

Reviewer #3 (Remarks to the Author):

The manuscript "Cryo-EM structure of the transcription termination factor Rho from *Mycobacterium tuberculosis* reveals mechanism of resistance to bicyclomycin" by Emmanuel Saridakis et al. targeted to characterize the atomic resolution structure of *M. tuberculosis* Rho [MtbRho] factor using cryo-electron microscopy and demonstrated the importance of mutation of leucine-to-methionine and effect of this mutation in BCM binding. The authors have successfully determined atomic resolution structures of the open conformation of MtbRho using cryo-EM. Additionally, the authors employed various biochemical assays to validate the binding of BCM.

This study is well-planned and executed appropriately. This study could attract a good amount of interest in this research field. However, some findings are not very clear or not represented accurately. Mainly most of the cryo-EM figures in the manuscript are not appropriately represented. My questions are listed below:

Major comments:

1. The authors started their manuscript about the structural characterization of MtbRho using cryo-EM. In supplemental Figure 2D, 2D class averages of negative staining images and cryo-EM images show the mixture of open-close form. However, the authors claimed only open conformation is visible in cryo-EM class averages. Some 2D class averages in supplemental figure 2D, e.g. class 9 (row 1, column 9), class 11 (row 2, column 1), and class 14 (row 2, column 4) appear as close form. Furthermore, the authors represented selected 2D class averages of MtbRho in the open state in figure 3B. Why do the authors see some close-open form of MtbRho in suppl figure 2D whereas only open conformation in supplemental figure 3B? Do the authors analyze all the cryo-EM class averages properly? It might be possible that the authors just selected those class averages where open conformation is visible in the original dataset (2,217,252). Therefore, they could not see any close conformation when they collected data using Titan Krios. Whereas when the dataset is small (JEOL data), authors considered all the particles and observed close-open both conformations of Rho. Therefore, the authors should explain why they observed some open-close conformations in negative staining/cryo-EM (Supp Fig 2D) and open-close conformations are completely missing in the main cryo-EM dataset (Supp Fig 3B).
2. In Figure 1B, the authors mentioned that "the ring gap (labeled with black dots)"; the black dots are not visible. Authors should represent the "ring gap" with a proper arrow or arrowheads. It is difficult for the readers to correlate the text with this figure. Authors could use enlarged views to demonstrate the "ring gap" and different orientations of the model for better visualization.
3. The authors superimposed six protomers to show the variability of the side chain Glu280. However, it is not clear from figure 2A. Authors claimed that "clear electron density in all the ATPase ...the phosphate, sugar, and adenine base moieties (Figures 1D, 2A&C)". However, there is no proper

labeling of phosphate, sugar, and adenine base moieties in Figure 1D or 2A, which is extremely difficult for the reader to correlate the text with figures.

4. Authors should present the cryo-EM map fitted with the final atomic model, representing overall fitting and good correlation with the map and the model. Additionally, authors should incorporate the EMRinger score for the model and map to show the quality of side-chain fitting in the models. The authors should add some additional images where side chains are clearly visible. Figure 1A shows that the EM map has a poorer resolution ($\sim 4.8 \text{ \AA}$) at protomer A and protomer F. How good is the fitting in these regions?

5. For Leu-Met mutation, the steric clash is not very clear from figure 3A. I have difficulties correlating the left-hand, middle, and right-hand panels of figure 3A.

6. The authors presented negative staining images and class averages in the manuscript. However, the authors did not mention anything about the methods of negative staining and class averages in EXPERIMENTAL PROCEDURES.

7. Data is collected $0.81 \text{ \AA}/\text{pix}$. However, initial classification is done at 320 pixel with $3.25 \text{ \AA}/\text{pix}$. If I consider data is binned by 4, pixel size will be $0.81 \times 4 = 3.24 \text{ \AA}/\text{pix}$. Again, data is reextracted at 400 pix box size but binned by $1.49 \text{ \AA}/\text{pix}$; generally, data is binned by 2 or 3 or 4. If data is binned by 2, the binned pixel should be 1.62, but the authors presented $1.49 \text{ \AA}/\text{pix}$. The author should explain properly how they achieve these pixel sizes.

8. How do authors generate the initial model?

9. In figure S4, the authors combined the 1st and 2nd classes together for final refinement. However, 2nd structure has an extra density in between protomers 2 and 4. Do the authors refine the 1st and 2nd models separately? It might be possible that 2nd model is showing some different conformation. Similarly, the 3rd model also shows some high-resolution features, 20% particles, and an extra density between protomers 1&2. The 3rd model might give another different conformation. Additionally, the authors classified 1,390,182 particles into only four classes. Authors should try to split the data into more classes.

Minor comments:

1. Figure 1A-C, please write the proper angle. It is not readable.

2. The author should mention "the SEC-MALS and SDS-PAGE gel experiment" instead of "SEC-MALS experiments (Figure S2A&B)" in the 1st line of the result and discussion.

3. Authors should use proper page numbers

4. Figure 1 has five figures, "A-E". However, in figure 1E, all the protomers are also labeled by A-F. It will be better to mark it differently (e.g. a-f or i-vi); otherwise, it will be difficult to read.

5. One high-res cryo-EM map fitted with the atomic model at different orientations should be presented.

6. It will be better to use a comma instead of the dot in figure S4 (e.g., 2,217,252). It is difficult to read.

We are grateful to the referees for their insightful reviews of our manuscript. Our detailed responses are provided in black below (page and reference numbers correspond to the revised manuscript).

Reviewer #1 (V. Nagaraja):

Saridakis et al in their manuscript describe the cryo-EM structure of M. tuberculosis Rho (MtbRho) at 3.3 Å resolution. Although it lacks the information on the unique N-terminal insertion domain of the protein which makes it different from Rho protein from well studied systems, it provides valuable insights into its co-factor preference, poor helicase activity and purine preference for ATPase activity. The structural insights into MtbRho provide an explanation for lower susceptibility to bicyclomycin (BCM). They also suggest that this reduced resistance to BCM is compensated by reduced motor efficiency of the protein. This is a thorough structural analysis of essential function from an important pathogen, further delineating the individual contribution of key residues. The experiments are for the most part well designed and the conclusions are supported by the data. I have a few remarks which authors may find useful if they desire to undertake the revision of the manuscript.

We thank the referee for his positive evaluation of our work and interesting remarks.

1. The comparison of Mtb Rho structure with that of E.coli revealed that most of BCM interacting residues are conserved. They hypothesize that the Met495 in MtbRho creates a bulk that can affect BCM binding by the steric clash. In vitro transcription assays using heterologous E.coli system were used to test this hypothesis. Would this strategy capture the species-specific signatures of termination and its inhibition given the differences between the species and their Rho?

Unfortunately, this question cannot be answered since nobody has performed Rho-dependent termination assays with mycobacterial RNA polymerase yet. All data published so far were with the same heterologous transcription assay used in the present study. However, this is not an issue when considering the specific question at hand, i.e. the role of Met495 in conferring resistance to BCM. The dramatic effect of the M495L mutation on BCM susceptibility (Fig. 3) strongly supports that the Leu-to-Met substitution in _{Mtb}Rho (as compared to _{Ec}Rho) is the main component of resistance to BCM. The structural data (e.g. BCM clash upon _{Ec}Rho:_{Mtb}Rho structural alignment) provide another convincing evidence.

2. M495L substitution in Mtb Rho converted it into more efficient ('E.coli like') using the heterologous transcription assay system. Unlike WT MtbRho, M495L substitution in the protein was able to unwind long RNA: DNA hybrids. As mentioned above an Mtb-based helicase activity assay system, using Mtb Rho-dependent terminator or RNA:DNA hybrids is likely to be more insightful given that Mtb Rho deals with high GC rich substrates.

It is indeed possible that using more genuine substrates could be informative but no mycobacterial *Rut* (Rho utilization) site has been identified so far and, thus, no genuine RNA:DNA hybrid mimic has been designed for use in helicase experiments. In a published study by the referee (PMID: 25229539), an RNA corresponding to the region downstream of the mycobacterial *sdaA* gene has been used in ATPase assays. However, this RNA was chosen based on *in silico* analysis (no predicted intrinsic terminator within it) and there is no experimental proof that it is indeed a physiological _{Mtb}Rho substrate. Moreover, this RNA is too long (330 nt) for PAGE-based helicase assays and, since its putative *Rut* site has not been delimited, it cannot be shortened and engineered readily into a suitable RNA:DNA substrate. In any case, using a standard RNA:DNA substrate (54.3%GC for the duplex part) we observed

a dramatic difference in helicase activity between WT _{Mtb}Rho and the M495L mutant (Fig. 4A), which is by itself sufficient proof of the impact of the M495 residue on the mechanochemistry of _{Mtb}Rho.

3. I would suggest the usage of term resistance to BCM is changed to lower susceptibility. From the data they presented in Fig 3B and C, BCM inhibits MtbRho albeit at higher concentrations. Other groups working on Mtb Rho have also reported these results. Use of the word resistance simply adds confusion in the literature.

It is a widely shared opinion that a lower susceptibility to a drug upon modification of the drug target is a common mechanism of acquisition of drug resistance (e.g., see PMID: 31294229 and references therein). For instance, single-point _{Ec}Rho mutations conferring resistance to BCM ‘only’ trigger 3-20 fold IC₅₀ effects (PMID: 10066795), similar to our case (Fig 3C). Similarly, as stated in PMID 15700959, “insusceptibility of single-point RNA polymerase mutants to rifamycins is not an all or nothing phenomenon”. In other words, we are certainly not denying the fact that susceptibility to BCM is lower rather than nonexistent, yet we consider the term “resistance” as adequate, widely accepted, and perhaps a clearer one in this situation.

4. From the structural analysis, it is clear that M495 is the main structural determinant of Mtb Rho introducing steric clash. They argue that hindrance could be with BCM itself or with the MtbLys359 side-chain from the Walker A motif (aka P-loop). To address these, BCM binding could be tested.

Indeed, BCM binding measurements would be useful but are very difficult if not impossible to perform reliably, due to the weak binding affinity of BCM. In the case of _{Ec}Rho, a K_d of 40-60 μM has been roughly estimated by microcalorimetry (PMID: 15725032). Unfortunately, no K_d could be determined for _{Mtb}Rho using the same method, probably because of an even lower affinity (PMID: 25999346).

Reviewer #2 (Anonymous):

Saridakis and colleagues report the first structure, obtained by cryo-EM, of the transcription termination factor Rho from a Gram-positive bacterium, namely Mycobacterium tuberculosis. This structure allows insights into the configuration of the Mtb-Rho hexamer, the organization of network required to bind ATP, and provided a hypothesis for the mechanism by which Mtb-Rho resists inactivation by bicyclomycin (BCM). This hypothesis, which predicted that a Leu/Met exchange at position 495 renders Mtb-Rho BCM resistant, was tested and validated biochemically. Interestingly, the M495L mutation causes BCM-resistance at the cost of enzymatic proficiency. Finally, the authors report on the evolutionary distribution of M495L-based BCM-resistance.

Rho is an attractive target for the development of new antibacterial agents, e.g. for the treatment of Tuberculosis. The lack of structural information on Rho from M. tuberculosis (or another gram-positive pathogen) has restricted these drug development efforts to somewhat unspecific phenotypic screen. This work thus addresses a major open question and I find it to be of high significance. It is unfortunate that the N-terminal insertion domain could not be resolved in this structure, but this is a minor weakness. Overall, this work is of very high quality and my suggestions are all minor:

We thank the reviewer for his/her kind words on our work.

1. Please check the labels in figure 1A. At least in my version, they seem incorrect and possibly incomplete.

Thank you for pointing this out. We had not noticed this software-based issue. This has now been corrected.

2. I find the discussion in the evolutionary tradeoff section somewhat ambiguous. Perhaps, it could be clarified if the authors want to suggest that the M495L mutation has been selected in mycobacteria specifically to mediate resistance to BCM. If so, it would be helpful to discuss what the evidence for this might be. Alternatively, the authors might want to add a few sentences on why this mutation occurs in mycobacteria if it was not selected to mediate BCM resistance.

This is a very interesting, kind of “egg-or-chicken-first” question but one for which we could only provide wild speculations. Nonetheless, to highlight the question we have slightly modified the manuscript page 11 as follows (modifications in light brown):

“Apart from Actinobacteria, we retrieved the Leu→Met mutation in two closely related insect endosymbionts from the Gram-negative Bacteroidetes phylum (Figure S14). This finding is surprising given the endosymbiont lifestyle, which itself should provide protection against BCM exposure. Although this suggests that the Leu→Met mutation has been acquired by chance or for another (unknown) reason than resistance to BCM, it is not clear yet if this conjecture can be extended to the case of Actinobacteria.

Overall, the Leu→Met mutation is rare and the only side-chain change observed at this position within the ~1300 representative Rho sequences compiled previously from multiple taxa⁴.”

Moreover, since mutations conferring antibiotic resistance often have a fitness cost (see PMID: 23554414), we realized that it is probably better/simpler to state “fitness cost associated to BCM resistance” than “evolutionary tradeoff between enzymatic efficiency and BCM resistance”. This avoids any not-solidly-grounded suggestion of the mutation being specifically selected to induce BCM resistance. The manuscript has been modified accordingly pages 11-12.

3. I don't expect the authors to add data on this (unless they chose to), but it would of course be interesting to know if a BCM-susceptible mutant of Mtb-Rho can replace the wt copy of Rho (in Mtb or M. smegmatis) and if this replacement makes mycobacteria BCM-sensitive.

Indeed, this would be very interesting (and might help answer point 2) but we do not have the expertise to perform this type of experiments.

Reviewer #3 (Anonymous):

The manuscript "Cryo-EM structure of the transcription termination factor Rho from Mycobacterium tuberculosis reveals mechanism of resistance to bicyclomycin" by Emmanuel Saridakis et al. targeted to characterize the atomic resolution structure of M. tuberculosis Rho [MtbRho] factor using cryo-electron microscopy and demonstrated the importance of mutation of leucine-to-methionine and effect of this mutation in BCM binding. The authors have successfully determined atomic resolution structures of the open conformation of MtbRho using cryo-EM. Additionally, the authors employed various biochemical assays to validate the binding of BCM. This study is well-planned and executed appropriately. This study could attract a good amount of interest in this research field. However, some

findings are not very clear or not represented accurately. Mainly most of the cryo-EM figures in the manuscript are not appropriately represented. My questions are listed below:

We thank the reviewer for his/her kind words and constructive comments that helped us improve the quality of our manuscript.

Major comments:

1. The authors started their manuscript about the structural characterization of MtbRho using cryo-EM. In supplemental Figure 2D, 2D class averages of negative staining images and cryo-EM images show the mixture of open-close form. However, the authors claimed only open conformation is visible in cryo-EM class averages. Some 2D class averages in supplemental figure 2D, e.g. class 9 (row 1, column 9), class 11 (row 2, column 1), and class 14 (row 2, column 4) appear as close form. Furthermore, the authors represented selected 2D class averages of MtbRho in the open state in figure 3B. Why do the authors see some close-open form of MtbRho in suppl figure 2D whereas only open conformation in supplemental figure 3B? Do the authors analyze all the cryo-EM class averages properly? It might be possible that the authors just selected those class averages where open conformation is visible in the original dataset (2,217,252). Therefore, they could not see any close conformation when they collected data using Titan Krios. Whereas when the dataset is small (JEOL data), authors considered all the particles and observed close-open both conformations of Rho. Therefore, the authors should explain why they observed some open-close conformations in negative staining/cryo-EM (Supp Fig 2D) and open-close conformations are completely missing in the main cryo-EM dataset (Supp Fig 3B).

It is a point that indeed needed to be clarified within the manuscript. From negatively stained M_{tb} Rho particle images (Figure S2C), we initially thought we were mostly observing closed M_{tb} Rho rings. By contrast, preliminary images of frozen-hydrated M_{tb} Rho particles recorded on our in-house cryo electron microscope actually revealed particles with mainly a U-like shape, consistent with an M_{tb} Rho open conformation. Upon closer examination, we realized that open particles were also present in the negatively stained M_{tb} Rho particle images. Corresponding 2D class averages also revealed that the intensity of subunits was not equivalent, which is consistent with the open corkscrew conformation of M_{tb} Rho. From images recorded using the Titan Krios microscope, we performed a 3D classification using two initial models (closed and open conformations derived from *E. coli* Rho atomic structures), asking for two 3D classes. This analysis confirmed that the M_{tb} Rho open conformation was predominant (79% of particles vs 21% for the closed conformation). The analysis of the closed-ring conformation did not allow us to compute a nice and reliable map whereas particles corresponding to the open conformation allowed computation of the final map at 3.3 Å resolution. To explain these points, we modified several sections of the manuscript. First, the Experimental procedure:

Image Processing and high-resolution 3D reconstruction

“A total of 10,888 movies were recorded. The frames of each movie were computationally dose-weighted and corrected for drift and beam-induced movement in RELION-3.1.037 using its own implementation. The contrast transfer function (CTF) of each micrograph was determined using Gctf-v1.18 program³⁸. A total of 2,217,252 particles were automatically picked with RELION’s autopicking procedure. Briefly, a template-free autopicking procedure based on a Laplacian-of-Gaussian (LoG) filter allowed to select and extract a set of particles from an initial small set of images and to compute corresponding 2D class averages. The best 2D class averages were then used as references in a second round of autopicking in order to optimize picking parameters. Once determined, the autopicking procedure was applied to all images. Particles were then sorted according to their correlation with the 2D class average used as reference, leading to discarding about 460,000 particle. The selected particle

images were then extracted with a box size of 320 X 320 pixels, binned to obtain a pixel size of 3.25 Å /pix, and submitted to 2D classification. Using the dedicated Relion procedure, the best 2D class averages allowed computation of an ab-initio model that was similar to the map computed from the atomic structure of the open form of E_c Rho (PDB 1PVO), confirming that the overall organization of M_{tb} Rho was similar to that of E_c Rho. Consequently, we performed a 3D classification using as initial models two density maps computed from the open and closed conformation of E_c Rho (PDB 1PVO and 2HT1). This analysis revealed that 79% and 21% of particles corresponded to the open and closed conformation of M_{tb} Rho respectively. The particles corresponding to the open form of M_{tb} Rho were subjected to a 3D classification with 4 classes using as initial model a map computed from the open E_c Rho structure. Two classes corresponding to 986,385 particles were selected and particles re-extracted with a box size of 400 X 400 pixels and binned at 1.49 Å /Pix. After two rounds of 3D auto-refinement without symmetry, a first density map at 3.97 Å was obtained. We then proceeded with per-particle defocus estimation, beam-tilt estimation plus Bayesian polishing. The final 3D refinement allowed us to compute a map at 3.32 Å resolution (FSC = 0.143) after post-processing. Figures were prepared using Chimera³⁹."

Supplementary figure 4 has been modified accordingly. We also modified the main text (pages 5 & 6):

"Images of negatively stained M_{tb} Rho complexes revealed, from 2D class averages, a hexameric organization of particles forming closed rings (Figure S2C) while these rings appear mostly open in conventional cryo-EM images (Figure S2D). Since some particles display an open conformation and the intensity of the M_{tb} Rho subunits is not uniform in negative stain images, we surmise that this discrepancy stems from a preferential orientation of the negatively stained particles onto the carbon film and a misalignment of hexameric particles during the 2D classification.

The most suitable conditions for high-resolution image acquisition in terms of M_{tb} Rho particles distribution and orientation in ice in presence of Mg-ATP and dC₂₀ ligands were obtained using Lacey grids. We recorded 10,888 movies of the M_{tb} Rho complexes using a Titan Krios instrument and processed them as described in Figures S3 and S4. This analysis revealed that the overall organization of M_{tb} Rho particles is similar to that of E_c Rho and that 79% and 21% of particles respectively correspond to open and closed ring organizations of the complex. Further analysis of the set of particles classified as "closed ring" revealed that about 92,000 particles with an open conformation were still present in this dataset. These misaligned particles, which do not display well-defined structural features, were not included in the final dataset. This, of course, means that the actual percentage of M_{tb} Rho particles in the closed ring conformation is even lower (16%). Moreover, the analysis of the closed ring dataset did not permit to compute a reliable and interpretable map, while particles corresponding to the open conformation allowed us to compute a cryo-EM map at 3.3 Å resolution (which led to the structure of the hexamer in the open ring conformation), showing an open-ring organization of the complex resulting from the assembly of six M_{tb} Rho subunits (Figure 1, Figure S5-7). Local resolution mapped on the structure with the RELION package ranges from 3.1 to 4.8 Å (Figure 1A-C). A gradient of high-to-low resolution is apparent in the 3D map from the innermost M_{tb} Rho ring subunits (labelled with stars in Figure 1B) to subunits at the ring gap (labeled with black dots). This gradient of resolution suggests some degree of flexibility among subunits, which was confirmed by a multibody analysis where the first main eigenvectors correspond to intersubunit twists (Figure S8). An atomic model of the M_{tb} Rho complex was built and refined based on the cryo-EM density map (Figure 1E, Table S1 and Figure S9) and is detailed below. Although the map resolution is not uniform, all ATP binding sites are resolved and the M_{tb} Rho atomic structure fits well even into the less well-resolved protomers of the cryo-EM map (Figures 1D and S5-S8)."

2. In Figure 1B, the authors mentioned that "the ring gap (labeled with black dots)"; the black dots are not visible. Authors should represent the "ring gap" with a proper arrow or arrowheads. It is difficult

for the readers to correlate the text with this figure. Authors could use enlarged views to demonstrate the "ring gap" and different orientations of the model for better visualization.

We fixed this software-based issue. The subunits are now clearly indicated with two black dots. Similarly, the best resolved subunits are indicated with black stars. We also added three figures in the supplementary material (S5, S6, S7), allowing the reader to have a better perception of the gap. These figures are mentioned in the revised version (first section of results; see point 1).

3. The authors superimposed six protomers to show the variability of the side chain Glu280. However, it is not clear from figure 2A. Authors claimed that "clear electron density in all the ATPase ...the phosphate, sugar, and adenine base moieties (Figures 1D, 2A&C)". However, there is no proper labeling of phosphate, sugar, and adenine base moieties in Figure 1D or 2A, which is extremely difficult for the reader to correlate the text with figures.

As stated in the text (page 7, line 160), the Glu280 residues are not resolved and thus cannot be shown on Fig 2A. We have modified the color-coding and annotations in Fig. 1D and Fig. 2A in order to help clarify the location of key components. Figure legends were modified accordingly. We anticipate that these modifications, together with the wealth of supplementary figures (old and new), will help readers grasp the salient features of the M_{tb} Rho structure.

4. Authors should present the cryo-EM map fitted with the final atomic model, representing overall fitting and good correlation with the map and the model. Additionally, authors should incorporate the EMRinger score for the model and map to show the quality of side-chain fitting in the models. The authors should add some additional images where side chains are clearly visible. Figure 1A shows that the EM map has a poorer resolution ($\sim 4.8 \text{ \AA}$) at protomer A and protomer F. How good is the fitting in these regions?

As mentioned in point 2, we have added three supplementary figures. Figure S5 shows a superimposition of the M_{tb} Rho atomic structure and the cryo-EM map. A few close-up views should help evaluate the quality of side-chain fitting. Figure S6 proposes a similar superimposition with close-up views for each subunit, which show that even for the less well-resolved subunits A and F, the fit remains rather good. Figure S7 focuses on the fit for the ATP sites. In addition, we have now calculated the EMRinger score with the Phenix suite, which is acceptable (1.29) and is now mentioned in the text (Page 17, line 385). The distribution of EMRinger peaks is provided here for further review, if needed:

5. For Leu-Met mutation, the steric clash is not very clear from figure 3A. I have difficulties correlating the left-hand, middle, and right-hand panels of figure 3A.

To better illustrate the steric clash, we have replaced the right-hand panel by a new view of the BCM binding cavity with E_c Rho and M_{tb} Rho superimposed. In the new panel, Bicyclomycin (sphere representation) invades the mesh surface of M_{tb} Rho-M495 but not that of E_c Rho-L320. We have also slightly modified the legend to clarify what is shown in Fig. 3A.

6. The authors presented negative staining images and class averages in the manuscript. However, the authors did not mention anything about the methods of negative staining and class averages in EXPERIMENTAL PROCEDURES.

Indeed, our mistake. We have now added a new section in the Supplementary information to describe the preliminary EM experiments performed to characterize the M_{tb} Rho-ATP-DNA complex:

Preliminary characterization of the M_{tb} Rho-ATP-DNA complex by negative stain

We checked the quality and homogeneity of the M_{tb} Rho-ATP-DNA complex sample by negative stain-electron microscopy. Three microliters of the M_{tb} Rho-ATP-DNA complex at 0.05 mg/ml were applied for 2 min on glow-discharged carbon-coated grids and then negatively stained with uranyl acetate 1 % for 1 min. Observation of EM grids was carried out on a JEOL 2200FS FEG Transmission Electron Microscope (TEM) operating at 200 kV under low-dose conditions (total dose of 20 electrons/Å²) in the zero-energy loss mode with a slit width of 20 eV. Images were recorded on a 4K × 4K slow-scan charge-coupled device camera (Gatan Inc.) at a nominal magnification of ×50,000 with defocus ranging from 0.5 to 1.0 μm. In total, 45 micrographs were recorded. The picking of particles was performed using e2boxer from Eman2 package⁸, and 2D class averages computed using RELION-3.1.0.

7. Data is collected 0.81 Å/pix. However, initial classification is done at 320 pixel with 3.25 Å/pix. If I consider data is binned by 4, pixel size will be 0.81 × 4 = 3.24 Å/pix. Again, data is reextracted at 400 pix box size but binned by 1.49 Å/pix; generally, data is binned by 2 or 3 or 4. If data is binned by 2, the binned pixel should be 1.62, but the authors presented 1.49 Å/pix. The author should explain properly how they achieve these pixel sizes.

According to the microscope calibration, the pixel size is in fact 0.8141 Å, explaining why after a binning of 4, we approximate the pixel size to 3.25 Å/pix. We modified the value of Pixel size in former Table S1 (now Table 1). For the second point, the binning is performed in Fourier space and not in real space. It is commonly used in cryo-EM image processing.

8. How do authors generate the initial model?

The way we generate the initial model is now included in the Experimental procedure “**Image Processing and high-resolution 3D reconstruction**” (see point 1 for the revised text of this section).

9. In figure S4, the authors combined the 1st and 2nd classes together for final refinement. However, 2nd structure has an extra density in between protomers 2 and 4. Do the authors refine the 1st and 2nd models separately? It might be possible that 2nd model is showing some different conformation. Similarly, the 3rd model also shows some high-resolution features, 20% particles, and an extra density between protomers 1&2. The 3rd model might give another different conformation. Additionally, the

authors classified 1,390,182 particles into only four classes. Authors should try to split the data into more classes.

In our process, the 3D classification was designed to produce 4 final 3D classes. The individual analysis of particles belonging to each 3D class produces 3D maps at lower resolution than the 3.3 Å resolution map proposed in this work. These maps did not display obvious structural differences between them. The extra-densities observed between some protomers never refined. They certainly correspond to additional residues present in $_{Mb}Rho$ and not present in $_{Ec}Rho$ but, whatever the analysis, these densities are never resolved. We also tried with 8 3D classes without any improvement, whether in terms of final resolution, the ability to resolve different conformation, or better resolving the central extra-densities.

Minor comments:

1. Figure 1A-C, please write the proper angle. It is not readable.

This has been corrected.

2. The author should mention "the SEC-MALS and SDS-PAGE gel experiment" instead of "SEC-MALS experiments (Figure S2A&B)" in the 1st line of the result and discussion.

The change has been made.

3. Authors should use proper page numbers

This has been corrected.

4. Figure 1 has five figures, "A-E". However, in figure 1E, all the protomers are also labeled by A-F. It will be better to mark it differently (e.g. a-f or i-vi); otherwise, it will be difficult to read.

To avoid confusion, we now use color labeling of the subunits in Fig. 1E but prefer to keep the same nomenclature as used for published structures of $_{Ec}Rho$, which facilitates comparison.

5. One high-res cryo-EM map fitted with the atomic model at different orientations should be presented.

Figures S5-S7 now show different orientations of the atomic model fitted into the cryo-EM map.

6. It will be better to use a comma instead of the dot in figure S4 (e.g., 2,217,252). It is difficult to read.

True, this has now been corrected.

Additonal changes made to the Ms:

-The E108 and R109 residues were mislabeled in Fig. 2B (E109, R108). This is now corrected.

- Required statements (reproducibility, data availability, etc.) have been added.

- As per journal instructions, the table on cryoEM statistics has been moved from the supplementary information (Table S1) to the main text (now Table 1).

REVIEWERS' COMMENTS:

Reviewer #1 (Remarks to the Author):

I had no major points in the original review and only a few clarifications and suggestions. The authors have addressed the points raised. I have no further questions.

Reviewer #2 (Remarks to the Author):

Excellent work. I have no further requests.

Reviewer #3 (Remarks to the Author):

Emmanuel Saridakis et al. modified the manuscript significantly and performed some extra experiments, like 3D classifications with close and open Rho models; and performed refinement, and calculated the EMringer score. Additionally, the authors have modified several areas of the manuscript and corrected the errors in the text and main figures. All the results and explanations are convincing. However, there are three minor comments/suggestions, which are listed below:

1. Most of the negative staining 2D class averages of MtbRho with Mg-ATP and dC20 appeared as close form. Generally, negative staining images are collected at dry condition (dehydrated condition), and most of the time, negative stain data are collapsed and flattening. I believe due to collapse and flattening issues, two protomers of Rho are approached closer and appear as close conformation. However, I agree with the authors that "2D class averages reveal that the intensity of protomers is not uniform", which might be due to the corkscrewed shape of the molecules.
2. In line 124 it is written, "79% and 21% of particles respectively correspond to open and closed ring ...". On the same page, line number 129 is written: "particles in the closed ring conformation are even lower (16%)". I think the authors want to say, "closed ring conformation are even lower (21%)". Please check it.
3. Supplementary Figure 6: Some loop regions of the protomer A (red protomer) is not fitted properly into the cryo-EM density map (see figure attached here, marked with arrows). The authors performed Phenix refinement, and it is expected that the atomic model should be inside the EM density map after Phenix refinement. This is the case for other protomers (B-F). It will improve the docking if the authors fit the protomer A separately into EM map and represent it the manuscript.

Reviewer #1:

I had no major points in the original review and only a few clarifications and suggestions. The authors have addressed the points raised. I have no further questions.

Reviewer #2:

Excellent work. I have no further requests.

Reviewer #3:

Emmanuel Saridakis et al. modified the manuscript significantly and performed some extra experiments, like 3D classifications with close and open Rho models; and performed refinement, and calculated the EMringer score. Additionally, the authors have modified several areas of the manuscript and corrected the errors in the text and main figures. All the results and explanations are convincing. However, there are three minor comments/suggestions, which are listed below:

1. Most of the negative staining 2D class averages of MtbRho with Mg-ATP and dC20 appeared as close form. Generally, negative staining images are collected at dry condition (dehydrated condition), and most of the time, negative stain data are collapsed and flattening. I believe due to collapse and flattening issues, two protomers of Rho are approached closer and appear as close conformation. However, I agree with the authors that “2D class averages reveal that the intensity of protomers is not uniform”, which might be due to the corkscrewed shape of the molecules.

We agree. This effect is also involved in this artifact. We integrated this possibility in the following sentence (page 6, lane 114):

« Since some particles display an open conformation and the intensity of the $M_{tb}Rho$ subunits is not uniform in negative stain images, we surmise that this discrepancy stems from a combination of several parameters such as the collapse and flattening of particles due to their drying during sample preparation, a preferential orientation of the negatively stained particles onto the carbon film, and a misalignment of hexameric particles during the 2D classification. »

2. In line 124 it is written, “79% and 21% of particles respectively correspond to open and closed ring ...”. On the same page, line number 129 is written: “particles in the closed ring conformation are even lower (16%)”. I think the authors want to say, “closed ring conformation are even lower (21%)”. Please check it.

Because of the artifact mentioned above, the overall percentage of $M_{tb}Rho$ truly in closed conformation is actually decreased. To clarify this, we have modified the sentence (page 6, lane 127):

« This, of course, means that the overall percentage of $M_{tb}Rho$ particles truly in the closed ring conformation is even lower (16%). »

3. Supplementary Figure 6: Some loop regions of the protomer A (red protomer) is not fitted properly into the cryo-EM density map (see figure attached here, marked with arrows). The authors performed Phenix refinement, and it is expected that the atomic model should be inside the EM density map after Phenix refinement. This is the case for other protomers (B-F). It will improve the docking if the authors fit the protomer A separately into EM map and represent it the manuscript.

This not due to a bad local fit but to a bad local resolution of the map. As shown in figure 1A-C, the local resolution map clearly shows that the two protomers A and F are the less well-resolved, displaying local resolution ranging from 3.5Å to 4.8Å. Protomer A was indeed docked separately (as was F) into the EM map at the initial stage of fitting. At this moderate local resolution range, however, the geometric constraints seem to (rightly) dominate the refinement. This explains why in some areas, the fit does not seem perfect, especially if the value of the threshold map is not suitable. This is commonly observed in cryo-EM with part of the map displaying that resolution range.

For Supplementary figures showing the $M_{tb}Rho$ cryo-EM map, we selected the most appropriate threshold for the best resolved protomers (C and D), allowing us to easily see high resolution details. Of course, this increases the visibility of problem areas in the less well-resolved part of the cryo-EM map.

We have now modified Supplementary figure 5 and 6, decreasing the threshold in order to reduce these visual effects.